# Estimating the gap between demand and supply of medical appointments by physicians for hypertension care: a pooled analysis in 191 countries

Rodrigo M Carrillo-Larco [1,2,3] Wilmer Cristobal Guzman-Vilca [2,4,5]
Dinesh Neupane [6]

For numbered affiliations see end of article.

**Correspondence to**
Dr Rodrigo M Carrillo-Larco;
rcarrill@ic.ac.uk

## ABSTRACT

**Introduction** With a growing number of people with hypertension, the limited number of physicians could not provide treatment to all patients. We quantified the gap between medical appointments available and needed for hypertension care, overall and in relation to hypertension treatment cascade metrics.

**Methods** Ecological descriptive analysis. We combined country-year-specific data on hypertension prevalence, awareness, treatment and control (from Non-Communicable Disease Risk Factor Collaboration) and number of physicians (from WHO). We estimated from 1 to 12 medical appointments per year for patients with hypertension. We assumed that physicians could see 25 patients per day, work 200 days/year and dedicate 10% of their time to hypertension care.

**Results** We studied 191 countries. Forty-one countries would not have enough physicians to provide at least one medical appointment per year to all the population with hypertension; these countries were low/lower middle income and in sub-Saharan Africa or East Asia and Pacific. Regardless of the world region, ≥50% of countries would not have enough physicians to provide ≥8 medical appointments to their population with hypertension. Countries where the demand exceeded the offer of medical appointments for hypertension care had worse hypertension diagnosis, treatment and control rates than countries where the demand did not exceed the offer. There were positive correlations between the physician density and hypertension diagnosis (r=0.70, p<0.001), treatment (r=0.70, p<0.001) and control (r=0.59, p<0.001).

**Conclusions** Where physicians are the only healthcare professionals allowed to prescribe antihypertensive medications, particularly in low and middle-income countries, the healthcare system may struggle to deliver antihypertensive treatment to patients with hypertension.

## INTRODUCTION

Globally, high blood pressure is the leading risk factor for cardiovascular morbidity and mortality.[1] Even though there are effective treatments for hypertension,[2] these do not always reach the patients leading to low hypertension treatment and control rates.[3][4]

### Strengths and limitations of this study

► We built on previous research which proposed a solid and reasonable methodology to quantify the gap between the *offer* and *demand* of medical appointments for hypertension care at the country level.

► All of our results are not for the same year. Potential country-specific users of our results are advised to understand our estimates taking into consideration the data year.

► Because of hypertension data availability, we restricted our analysis to the age group of 30–79 years.

► We used the number of physicians as reported by the WHO, and assumed each physician sees 25 patients per day and works 200 days/year. We are not aware of empirical data to support this (or other) scenario.

► Country-level estimates often hide within country differences.

Reasons for poor access to antihypertensive treatment could include lack of medicines, inability to afford treatment and not having health insurance. However, where universal health coverage is available, the obstacle to provide treatment for patients with hypertension could be the shortage of physicians,[5] particularly where physicians are the only health professionals authorised to prescribe medicines for hypertension. This may be the case in many countries, though there are no empirical data to characterise the actual organisation of hypertension care in all countries. Due to the lifelong nature of antihypertensive treatment, people with hypertension will need multiple medical appointments per year whereby physicians will titrate and refill their medicines. While the number of people with hypertension grows faster than the number of physicians, and hypertension diagnosis rates increase,[4] countries will struggle to provide the medical appointments

needed for patients with hypertension; that is, there will be more medical appointments needed than available for hypertension care.

The gap between the number of medical appointments needed and available for people with hypertension has been quantified showing that several countries do not have enough physicians for 3, 6 or 12 medical appointments per year for people with hypertension.[6] Nonetheless, previous estimates were up to 2015 and the authors did not study the gap in relation to hypertension care cascade metrics: diagnosis, treatment and control. Using more recent data will provide countries with up-to-date evidence to inform their policies, interventions and targets. Also, the correlations between the gap and hypertension care cascade metrics will inform if countries with larger gaps also have worse hypertension treatment and control rates.

Consequently, elaborating on previous evidence and overcoming the above described limitations,[6] we quantified the gap between the number of medical appointments needed by people with hypertension and the number of medical appointments available for hypertension care based on the latest available data.

## METHODS

This is an ecological descriptive analysis. We combined data from different sources to quantify the hypertension gap regarding *demand* of medical appointments by people with hypertension and the availability of physicians to provide medical appointments for hypertension care (ie, *offer*). In addition, we presented the gap estimates in relation to hypertension treatment cascade metrics: diagnosed, treated and controlled. Further details about the data sources and analysis are shown in online supplemental file 1.

### Data sources

The prevalence of hypertension in each country among people aged 30–79 years was retrieved from the Non-Communicable Disease Risk Factor Collaboration (NCD-RisC)[4 7]; the data are publicly available online.[7] From this data source we also extracted the proportion diagnosed, treated and controlled among people with hypertension (ie, hypertension care cascade).[4] The NCD-RisC estimates are sex stratified. In here, we reported one estimate per country (ie, men and women combined). For this, we multiplied the sex-specific prevalence by the number of men and women, and summed these products to have the country prevalence.

The number of physicians in each country was retrieved from the WHO.[8] Population data (number of people per country) were retrieved from the Institute for Health Metrics and Evaluation ().[9] The world region and income group for each country was retrieved from the World Bank.[10]

The hypertension data (source: NCD-RisC[4 7]) are available for all years until 2019, whereas the number of physicians for each country is available at different years (ie, some countries have more recent data than others). We combined the number of physicians with the year-corresponding hypertension data (eg, number of physician data in 2017 with hypertension data in 2017); population data were for the same year. We only used the most recent World Bank region and income classification (regardless of the physician or hypertension data year).[10]

### Variables

In here, *demand* refers to the number of medical appointments needed by people with hypertension. As in a previous paper,[6] we assumed three scenarios: 3, 6 and 12 medical appointments by year. We multiplied the absolute number of people with hypertension by 3, 6 and 12 to compute the number of medical appointments needed. In addition to these three scenarios, we considered 1 through 12 medical appointments per year; that is, if people with hypertension would need from 1 to 12 medical appointments per year.

In here, *offer* refers to the response of the health system to the appointments needed (*demand*) by people with hypertension. We assumed that a physician could see 25 patients per day and he/she works 200 days/year (25×200=5000); we multiplied the number of physicians by 5000 to compute the number of medical appointments available. This product was further multiplied by 0.10 because we assumed that 10% of a physician's time is dedicated to hypertension care.[6]

In here, *the gap* refers to the difference between the *offer* and the *demand*. That is, we subtracted the absolute number of medical appointments needed (*demand*) from the absolute number of medical appointments available (*offer*) for the three scenarios (3, 6 and 12 medical appointments) and from 1 through 12 medical appointments per year. These subtractions were further divided by 1 000 000 so that the *gap* is reported per 1 000 000 medical appointments. A negative gap suggests that the *demand* exceeded the *offer*; that is, there were more medical appointments needed than available for hypertension care.[6]

### Statistical analysis

Results are presented as counts (absolute numbers), as proportions or percentages, as well as medians. Heatmaps and line plots are used to show at what number of medical appointments needed the gap turned negative. We used world maps to show countries with a negative gap according to the three scenarios (3, 6 and 12 medical appointments). We used boxplots to depict the mean of the hypertension care cascade metrics (diagnosed, treated and controlled) by the three scenarios among countries with positive and negative gaps. We used scatterplots to correlate the physician density (absolute number of physicians per 1000 people) and the hypertension care cascade metrics; these plots had the Pearson correlation coefficients.

## Patient and public involvement

It was not appropriate or possible to involve patients or the public in the design, or conduct, or reporting, or dissemination plans of our research.

## RESULTS

### Data characteristics

There were 191 countries: 30 in East Asia and Pacific, 49 in Europe and Central Asia, 33 in Latin America and the Caribbean, 20 in Middle East and North Africa, 2 in North America, 8 in South Asia and 47 in sub-Saharan Africa; of note, two countries were not classified into any of these groups: Cook Islands and Nauru. Fifty-eight countries were high income, 54 were upper middle income, 49 were lower middle income and 28 were low income; of note, two countries (Cook Islands and Nauru) were not classified into any of these groups. Data regarding the number of physicians were available from 2004 to 2019, with 85% available since 2015.

### Underlying variables

The prevalence of hypertension ranged from 20% to 56%; the median across all countries was 37%; the median by region was 42% in Europe and Central Asia, 40% in Latin America and the Caribbean, 36% in South Asia, 35% in East Asia and Pacific, 34% in sub-Saharan Africa, 33% in Middle East and North Africa and 32% in North America.

The density of physicians (per 1000 people) ranged from 0.07 to 13.77 (median=3.26); the median physician density (per 1000 people) according to the region was 0.36 in sub-Saharan Africa, 1.75 in East Asia and Pacific, 1.99 in South Asia, 3.67 in Middle East and North Africa, 3.84 in Latin America and the Caribbean, 4.34 in North America and 6.09 in Europe and Central Asia.

### Gap: available (*offer*) medical appointments minus demand

Out of 191 countries in the analysis, 41 would have a negative gap to provide one medical appointment per year for people with hypertension (figure 1); that is, in 41 countries, and under the assumptions of our analysis, the healthcare system would not have enough physicians for all patients with hypertension to have at least one medical appointment per year.

According to their income, these 41 countries were either low income (21 countries) or lower middle income (19 countries); one country (Botswana) was upper middle income. According to their location, these 41 countries were in sub-Saharan Africa (35 countries; largest gap in Democratic Republic of the Congo), East Asia and Pacific (4 countries; largest gap in Papua New Guinea), Middle East and North Africa (1 country; Djibouti) and Latin America and the Caribbean (1 country; Haiti).

The number of countries with a negative gap (more *demand* than *offer*) increased with the number of medical appointments per year (figure 2). Africa had the largest proportion of countries with a negative gap regardless of the number of medical appointments. Regardless of the

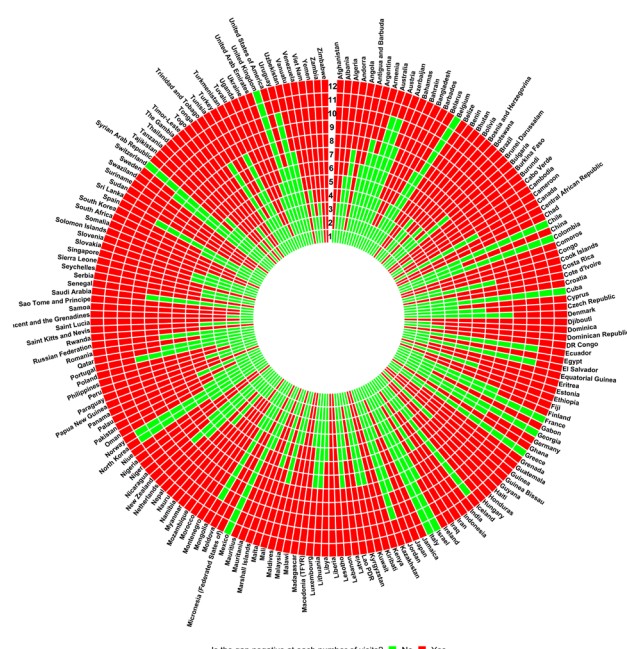

**Figure 1** Indicator of positive or negative gap by number of medical appointments (1 through 12) per year by country. A negative gap indicates that there is more demand (need of medical appointment by people with hypertension) than offer (number of physicians).

world region, ≥50% of the countries could not offer eight or more medical appointments per year to the population with hypertension.

### Gap: available (*offer*) medical appointments minus demand in fixed scenarios

For a fixed number of three medical appointments (ie, every 4 months) per year, 80 (42%) countries would have a negative gap (more *demand* than *offer*); 24 of these countries were low income whereas two were high income (figure 3). For a fixed number of six medical appointments (ie, twice per year) per year, 132 (69%) countries would have a negative gap; 27 of these countries were low income while 23 were high income. For a fixed number of 12 medical appointments (ie, monthly) per year, 179 (94%) countries would have a negative gap; 28 of these countries were low income while 50 were high income.

### Gap: available medical appointments minus demand in fixed scenarios and in relation to hypertension diagnosis, treatment and control

After stratifying the number of visits according to the three scenarios (3, 6 and 12 medical appointments per year), those countries with a negative gap (more *demand* than *offer*) had worse hypertension treatment cascade metrics than those countries without a negative gap (figure 4). For example, across countries with always a negative gap, the median prevalence of diagnosed hypertension was 40% versus a median prevalence of 63% across countries without a negative gap; the corresponding estimates for hypertension treatment were 23% vs 52%, and for hypertension control were 9% vs 23%. That is, in countries

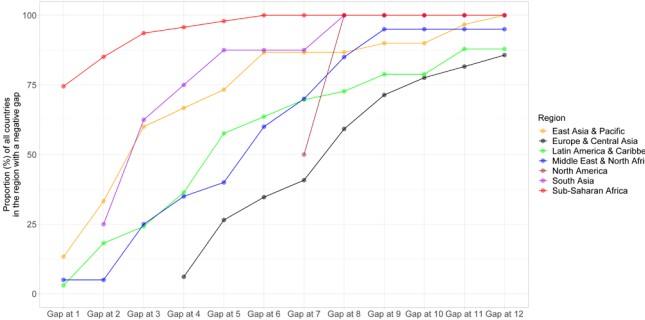

**Figure 2** Proportion of countries with a negative gap by number of medical appointments (1 through 12) per year by world region. Two countries (Cook Islands and Nauru) were not included in the plot because their region was not available in the data source (World Bank).

where the *demand* exceeded the *offer*, the hypertension treatment cascade metrics were worse (on average) than in countries where the *demand* did not exceed the *offer* of medical appointments for hypertension care per year.

### Correlation between hypertension diagnosis, treatment and control with physician density

For all three hypertension care cascade metrics, there was a positive correlation with the physician density (figure 5). The magnitude of the correlation was stronger for hypertension diagnosis and treatment than for hypertension control. Low-income countries were most often found in the bottom left quadrant, with worse hypertension care cascade metrics and lower physician density. Conversely, high-income countries were often found in the upper right quadrant, with better hypertension care cascade metrics and higher physician density.

### DISCUSSION

Leveraging on high-quality and official data sources[7 9 10] and building on previous research,[6] we documented that 41 countries would not have enough physicians to provide at least one medical appointment per year to their

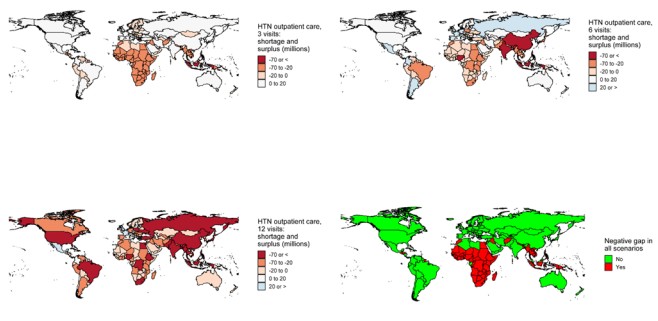

**Figure 3** World maps showing countries according to the gap level by the three possible scenarios, and signalling which countries would always have a negative gap regardless of the scenario. HTN: hypertension.

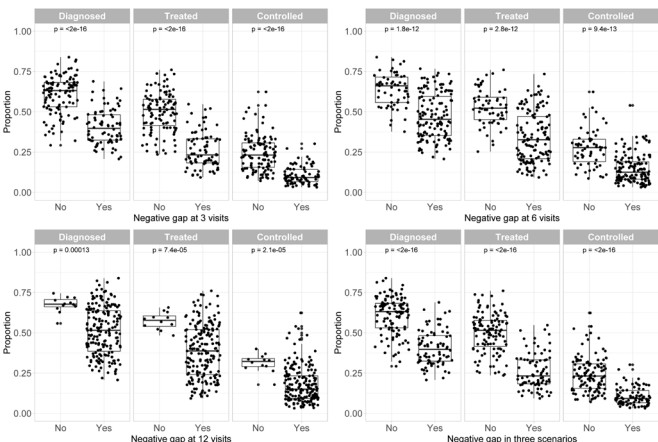

**Figure 4** Hypertension treatment cascade metrics by positive or negative gap stratified by the three possible scenarios. Each dot represents a country.

population with hypertension; these countries were either low or lower middle income and located in sub-Saharan Africa or in East Asia and Pacific. Regardless of the world region, more than half of all countries would not have enough physicians to provide eight or more medical appointments to their population with hypertension per year. On average, countries where the *demand* exceeded the *offer* of medical appointments for people with hypertension also had worse hypertension care cascade metrics than those countries where the *demand* did not exceed the *offer*. There were positive correlations between the density of physicians and hypertension diagnosis, treatment and control rates.

The health consequences of these gaps would include fewer opportunities to access medical appointments for hypertension control, hence higher likelihood to not reach optimal blood pressure levels through pharmacological treatment. This is partially supported by the ecological correlations we reported, whereby higher physician density was positively correlated with higher rates of hypertension treatment and control. Further from not achieving hypertension control, this could lead to higher morbidity, mortality and disability due to cardiovascular diseases.

We followed the methodology developed by the authors of the only global work on this subject.[6] Despite minor changes in our methodology (eg, updated data sources) which led to different numeric results, we reached similar conclusions[6]: many low and middle-income countries would struggle to provide medical appointments per year to their population with hypertension. This does not mean that the more medical appoints for hypertension care the better; in other words, quantity does not necessarily reflect quality. We advocate for innovative approaches whereby people with hypertension receive the best evidence-based care available, so that they receive pharmacological and non-pharmacological care to delay or avoid complications.

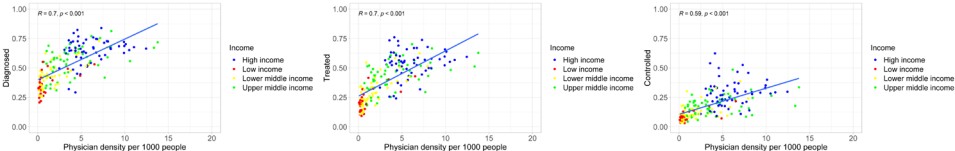

**Figure 5** Correlation between hypertension treatment cascade metrics and physician density. Each dot is a country; colours indicate the income group of each country.

Vedanthan *et al* reported a positive association between physician density and hypertension treatment rates in the unadjusted model only[5]; potential explanations for the lack of statistical significance in adjusted model include data collation year and not having estimates for some countries where there could have been a large gap. We computed correlation coefficients showing a positive correlation between physician density and hypertension treatment; arguably, this shows a similar result to their unadjusted estimates.[5] However, we advanced the literature by showing the correlation between physician density and hypertension diagnosis and control.

We built on previous research which proposed a solid and reasonable methodology to quantify the gap between the *offer* and *demand* of medical appointments for hypertension care at the country level.[6] This work substantially advanced our previous publication by using the most recent hypertension data which are comparable across all countries[4]; by focusing on hypertension (rather than raised blood pressure[6]), which accounts for aware and unaware cases; and by studying the gap between *offer* and *demand* in relation to hypertension care cascade metrics.

Notwithstanding, there are limitations we need to acknowledge. First, all of our results are not for the same year. Potential country-specific users of our results are advised to understand our estimates taking into consideration the data year (online supplemental file 2). Health organisations should also make efforts to secure consistent, comparable and recent data about the number of physicians in all countries and when possible, stratified by medical specialty. Second, because of hypertension data availability, we restricted our analysis to the age group 30–79 years.[4] Perhaps, older people may require more medical appointments because of multimorbidity and polypharmacy. If so, our results show a conservative scenario and alert that the gap between *offer* and *demand* of medical appointments for hypertension care could be larger if we had included the entire adult population. Third, we used the number of physicians as reported by the WHO, and assumed each physician sees 25 patients per day and works 200 days/year.[6] Although conservative and possible, we are not aware of empirical data to support this (or other) scenario. Any other scenario, either to replace our assumption or as a sensitivity analysis, would also be an educated assumption because of the lack of empirical evidence to inform the scenarios. International health organisations, through surveys or country consultations, could collect data on the actual load of the healthcare workforce and how much time is allocated to key tasks (eg, non-communicable diseases care). Fourth, country-level estimates often hide within country differences. Our results should be interpreted in light of this limitation, and we encourage researchers and health officials to conduct similar analysis at the subnational level to tailor local policies and interventions. Fifth, the correlations between the gap and the hypertension care cascade metrics were at the country level. This ecological analysis should be cautiously interpreted and need further verification with individual-level data. Sixth, we conducted a descriptive work to characterise the gap between medical appointments by physicians needed and available for people with hypertension worldwide. A quantification of the health consequences of this gap was beyond the scope of this work, though it is undoubtedly a relevant question that deserves attention. Finally, the most salient limitation of our work is that we assumed only physicians prescribe treatment for hypertension yet other healthcare personnel may participate in this task. Unfortunately, there are no global data to make the distinction on whether all hypertension care is provided by physicians or other personnel. In other words, there are no global data (eg, encyclopaedia) characterising the actual organisation of hypertension care in all countries. This descriptive work aimed to quantify the gap at the country level, and providing an in-depth profile of the organisation of hypertension care in 191 countries is beyond the scope of this work or any available global data. This work alerts of a large gap to meet the needs of people with hypertension, though the gap may be smaller if other healthcare personnel provide treatment for people with hypertension.

The fact that 41 countries would not have enough physicians to provide one medical appointment per year to their population with hypertension is alarming; similarly, having three medical appoints per year (ie, one every 4 months) was also unfeasible in many countries.

Task-shifting strategies[11 12] could offer a potential solution for countries where the *demand* exceeded the *offer* of medical appointments for hypertension care. This could include training nurses and other non-physician health professionals,[13–16] as well as community health workers or champions of the community,[16] to monitor blood pressure and to dispense antihypertensive treatment. Physicians could oversee these strategies and focus on difficult cases, such as patients with resistant hypertension or when treatment requires careful tailoring. Having antihypertensive treatment refilled in the local pharmacy could also help.[16 17] Future research should quantify the impact

of task-shifting strategies to improve care for people with hypertension.

Countries could adopt prescription strategies that meet their resources. For example, if a country could only have four medical appointments per year for people with hypertension, they could provide medicines for 3 months. Patients would have enough pills until the next appointment. This would require careful consideration by the physician to identify patients who would not need frequent medical appointments, according to the patient's overall health profile.

There is strong evidence showing the value of non-physician health professionals to improve hypertension care.[11 12 16 17] Selecting the best examples which could be implemented following rigorous research methods and evaluations deserves prompt attention.

The COVID-19 pandemic would impact our results and the associated implications. First, the latest physician data were for 2019. Unfortunately, many physicians have recently lost their lives to COVID-19,[18] potentially widening the herein documented gaps especially in low and middle-income countries. Second, COVID-19 disrupted health services because they stopped their regular activities to take care of patients with COVID-19; lockdowns also disrupted regular health services.[19] Therefore, people with chronic diseases including hypertension had their usual care interrupted. This could have led to treatment discontinuation and fewer patients meeting blood pressure targets. As health services resume their regular activities, patients with chronic conditions may need more than usual care to catch up with their treatments. This would translate into more frequent medical appointments widening the gap herein documented.

Brain drain of physicians could also play a relevant role.[20] Physicians from low and middle-income countries, where we found the largest gaps, tend to migrate to high-income countries. If this continues, the herein document gaps in low and middle-income countries would increase.

Both of these potential drivers could be solved with more physicians, though this may not be a realistic option. To support timely and continuous treatment for people with hypertension where physicians are the only health professional authorised to prescribe medicines, it seems imperative to find solutions that do not require physicians only.

Many low and middle-income countries would not meet the *demand* of medical appointments needed by people with hypertension. This would lead to lower rates of hypertension treatment and control at the country level. Because hypertension is the leading risk factor for cardiovascular morbidity and mortality,[1] and has poor treatment and control rates in many low and middle-income countries,[4] healthcare systems should secure better care for people with hypertension including the uninterrupted provision of antihypertensive medication. A synergic participation of physicians, non-physician healthcare workers and lay people (eg, community health workers) would be needed to secure continuous treatment for (all) people with hypertension.

**Author affiliations**
¹Department of Epidemiology and Biostatistics, School of Public Health, Imperial College London, London, UK
²CRONICAS Centre of Excellence in Chronic Diseases, Universidad Peruana Cayetano Heredia, Lima, Peru
³Universidad Continental, Lima, Peru
⁴School of Medicine "Alberto Hurtado", Universidad Peruana Cayetano Heredia, Lima, Peru
⁵Sociedad Científica de Estudiantes de Medicina Cayetano Heredia (SOCEMCH), Universidad Peruana Cayetano Heredia, Lima, Peru
⁶Department of International Health, Johns Hopkins Bloomberg School of Public Health, Johns Hopkins University, Baltimore, Maryland, USA

**Contributors** RMC-L, WCG-V and DN conceived the study. RMC-L, WCG-V and DN discussed the planning and execution of the study. WCG-V collated, pooled and conducted the analysis. RMC-L conducted the analysis and drafted the first version of the manuscript. RMC-L, WCG-V and DN discussed the findings and provided critical input to the manuscript. RMC-L, WCG-V and DN approved the submitted version. RMC-L submitted the work for publication. RMC-L, WCG-V and DN addressed the comments by the external peer reviewers. RMC-L and WCG-V accept full responsibility for the work and the conduct of the study, had access to the data, and controlled the decision to publish.

**Funding** RMC-L is supported by a Wellcome Trust International Training Fellowship (Wellcome Trust 214185/Z/18/Z). DN receives support from Resolve to Save Lives, an initiative of Vital Strategies, which is funded by Bloomberg Philanthropies, the Bill & Melinda Gates Foundation and Gates Philanthropy Partners, which is funded with support from the Chan Zuckerberg Initiative.

**Disclaimer** The opinions herein expressed are those of the authors alone. The funders had no role in the study design, analysis, results interpretation or conclusions.

**Map disclaimer** The inclusion of any map (including the depiction of any boundaries therein), or of any geographic or locational reference, does not imply the expression of any opinion whatsoever on the part of BMJ concerning the legal status of any country, territory, jurisdiction or area or of its authorities. Any such expression remains solely that of the relevant source and is not endorsed by BMJ. Maps are provided without any warranty of any kind, either express or implied.

**Competing interests** None declared.

**Patient and public involvement** Patients and/or the public were not involved in the design, or conduct, or reporting, or dissemination plans of this research.

**Patient consent for publication** Not required.

**Ethics approval** We used data available in the public domain and at the country level. The data did not contain any personal identifiers. We did not seek approval by an ethics committee.

**Provenance and peer review** Not commissioned; externally peer reviewed.

**Data availability statement** All data relevant to the study are included in the article or uploaded as supplementary information. All data sources herein analysed are in the public domain, and links to these sources have been provided in the main manuscript. We are also providing the analysis data (ie, after pooling individual data sources) as online supplemental file 3.

and indication of whether changes were made. See: https://creativecommons.org/licenses/by/4.0/.

**ORCID iDs**
Rodrigo M Carrillo-Larco http://orcid.org/0000-0002-2090-1856
Wilmer Cristobal Guzman-Vilca http://orcid.org/0000-0002-2194-8496
Dinesh Neupane http://orcid.org/0000-0002-1501-2990

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
