## [Reviewer comments · BMJ Open]

ARTICLE DETAILS

TITLE (PROVISIONAL)	Estimating the gap between demand and supply of medical appointments by physicians for hypertension care: A pooled analysis in 191 countries
AUTHORS	Carrillo Larco, Rodrigo M.; Guzman-Vilca, Wilmer Cristobal; Neupane, Dinesh

VERSION 1 – REVIEW

REVIEWER	Wang, Ji Shanghai Jiao Tong University Medical School Affiliated Ruijin Hospital
REVIEW RETURNED	07-Jan-2022

GENERAL COMMENTS	The work presented in this manuscript is relevant for global health. There are several suggestions for revision of the manuscript. 1. The authors may consider to provide a summary table to categorize the countries according to the size of the gap, and to present the demand, offer, gap, the population size and the estimated health consequences because of the gap. Such information would substantiate with numbers that it is truly a global health problem, and the problem is a big one.2. The characters in some of the figures are not readable. The authors may need to improve clarity of these figures, especially Figures 2 and 4.3. In the Discussion, the authors may consider to touch a little bit the possible strategies to deal with the problem. In low- and middle-income countries, it is probably not possible to fill the gap by any means in the foreseeable future. What can these countries do ? Wait and see ? No. Raising questions is important. Finding solutions is more important.
---

REVIEWER	Wouters, Eveline Tilburg University, Tranzo, School of Social and Behavioral Sciences
REVIEW RETURNED	07-Jan-2022

GENERAL COMMENTS	Gap between demand and offer of medical appointments for hypertension care: A global analysis This is a study that aims to quantify the gap between the number of medical appointments needed by people with hypertension and
---

	the number of medical appointments available for hypertension care based on the latest available data. This contributes to our understanding that health care as it is currently organised, will not hold in future to provide basic care for hypertension patients. As such, this is an interesting study. Nevertheless, I would like to address some point of concern: (1) General:  - Marked yellow in the pdf, some textual inconsistencies/ remarks - Keywords: see remark in pdf (2) Main: Introduction: I would like to have some more information as to how hypertension treatment is organised worldwide. The assumption in the paper is that physicians are mandatory and (physical?) appointments necessary, as in some countries this is, due to shortage of physicians, otherwise organised. E.g. by nurse specialists / physician assistants or with support of eHealth. This is lacking now, although this is the main conclusion and practical implication of the discussion. Therefore, the paper would benefit from more information about the actual organisation of hypertension care in the countries analysed.
--	--

VERSION 1 – AUTHOR RESPONSE

REVIEWER #1

We appreciate the positive feedback by Dr Wang. We did our best to address all his suggestions.

Q1. The authors may consider to provide a summary table to categorize the countries according to the size of the gap, and to present the demand, offer, gap, the population size and the estimated health consequences because of the gap. Such information would substantiate with numbers that it is truly a global health problem, and the problem is a big one.

R1. We are providing a new Supplementary Table 1. In the original submission, Supplementary Table 1 only showed the gap. In this new Supplementary Table 1, as suggested by the reviewer, we are reporting the demand, offer, sample size as well as the gap. Because we have results for 191 countries, and different gaps based on the number of medical appointments (from one through 12), this table is very busy and perhaps utterly overwhelming for the main text. This is the reason why we included this table as Supplementary Material. If the editors believe the table would fit nicely in the main text, we would be happy to move it to the main text.

Reporting on the health consequence for each country is, unfortunately, beyond the scope of our work. Estimating the health consequence of the gaps herein quantified would require a different study design, other data sources, and should be a paper on its own. We acknowledged this limitation in the Discussion section (p. 11): “Sixth, we conducted a descriptive work to characterize the gap between medical appointments by physicians needed and available for people with hypertension worldwide. A quantification of the health consequences of this gap was beyond the scope of this work, though it is undoubtedly a relevant question that deserves attention.”

We also briefly commented on the consequences of these gaps in the Discussion section. The new paragraph reads (p. 09): “The health consequences of these gaps would include fewer opportunities to access medical appointments for hypertension control, hence higher likelihood to not reach optimal blood pressure levels through pharmacological treatment. This is partially supported by the ecological correlations we reported, whereby higher physician density was positively correlated

with higher rates of hypertension treatment and control. Further from not achieving hypertension control, this could lead to higher morbidity, mortality and disability due to cardiovascular diseases.”

Q2. The characters in some of the figures are not readable. The authors may need to improve clarity of these figures, especially Figures 2 and 4.

A2. All the figures were improved, particularly Figure 2 and Figure 4 as recommended. If our manuscript is accepted for publication, we will be happy to work with the editors and typesetters to improve the figures so that they meet the standards of the journal.

Q3. In the Discussion, the authors may consider to touch a little bit the possible strategies to deal with the problem. In low- and middle-income countries, it is probably not possible to fill the gap by any means in the foreseeable future. What can these countries do? Wait and see? No. Raising questions is important. Finding solutions is more important.

A3. We briefly discussed the health consequences of the gaps; kindly refer to our first answer in which we signal two new paragraphs incorporated in the manuscript (p. 09 and p. 11; Discussion section).

Potential strategies and solutions were also discussed. On page 11 we elaborated on “task shifting strategies” as a potential solution whereby hypertension care is addressed by a team of physicians, nurses, dieticians and other healthcare personnel rather than the physician alone. On page 12 we elaborated on potential solutions to improve treatment allocation, including, for example, three-month refill (rather than monthly refill) of antihypertensive medication and to include pharmacists in this task (in the context of task shifting strategies as discussed on page 11).

REVIEWER #2

We appreciate Dr Wouters took the time to comment on our work. We have incorporated her suggestions.

Q1. Marked yellow in the pdf, some textual inconsistencies/ remarks.

A1. We have comprehensively proofread our manuscript. We feel confident these concerns have been addressed already. Thank you very much for your comprehensive review.

Q2. Keywords: see remark in pdf.

A2. We included two more keywords (underlined): “non-communicable diseases; population health; health metrics; health systems; hypertension”. The last keyword was suggested by the reviewer.

Q3. Main: Introduction: I would like to have some more information as to how hypertension treatment is organised worldwide. The assumption in the paper is that physicians are mandatory and (physical?) appointments necessary, as in some countries this is, due to shortage of physicians, otherwise organised. E.g. by nurse specialists/physician assistants or with support of eHealth. This is lacking now, although this is the main conclusion and practical implication of the discussion. Therefore, the paper would benefit from more information about the actual organisation of hypertension care in the countries analysed.

A3. A profile of the actual organization of hypertension care in the 191 countries included in the analysis would be valuable, yet unfortunately this is not possible at this time. To the best of our knowledge, there is no consistent or comparable data to characterize the organization of hypertension care in all countries. Doing so would require, at least, a systematic search of all grey and scientific literature to understand the organization, services and resources in each health system of 191 countries. Notably, some countries would have more than one health system (e.g., public vs private), making this task even more difficult. We hope the reviewer will bear with us on this limitation, and kindly acknowledge this request is beyond the scope of our work. We have, however, addressed this as the main limitation of our work (p. 11; new text underlined): “Finally, the most salient limitation of our work is that we assumed only physicians prescribe treatment for hypertension yet other healthcare personnel may participate in this task. Unfortunately, there are no global data to make the distinction on whether all hypertension care is provided by physicians or other personnel. In other words, there is no global data (e.g., encyclopaedia) characterizing the actual organization of hypertension care in all countries. This descriptive work aimed to quantify the gap at the country level, and providing an in-depth profile of the organization of hypertension care in 191 countries is beyond the scope of this work or any available global data.”

This subject was also acknowledged in the Introduction section (p. 04; new text underlined): “...the obstacle to provide treatment for patients with hypertension could be the shortage of physicians, particularly where physicians are the only health professionals authorized to prescribe medicines for hypertension. This may be the case in many countries, though there is no empirical data to characterize the actual organization of hypertension care in all countries.”

VERSION 2 – REVIEW

REVIEWER	Wang, Ji Shanghai Jiao Tong University Medical School Affiliated Ruijin Hospital
REVIEW RETURNED	18-Feb-2022
GENERAL COMMENTS	No further comments.